# Reconciling Security and Communication Efficiency in Federated Learning

## Abstract

Cross-device Federated Learning is an increasingly popular machine learning setting to train a model by leveraging a large population of client devices with high privacy and security guarantees. However, communication efficiency remains a major bottleneck when scaling federated learning to production environments, particularly due to bandwidth constraints during uplink communication. In this paper, we formalize and address the problem of compressing client-to-server model updates under the Secure Aggregation primitive, a core component of Federated Learning pipelines that allows the server to aggregate the client updates without accessing them individually. In particular, we adapt standard scalar quantization and pruning methods to Secure Aggregation and propose Secure Indexing, a variant of Secure Aggregation that supports quantization for extreme compression. We establish state-of-the-art results on LEAF benchmarks in a secure Federated Learning setup with up to $40\times$ compression in uplink communication and no meaningful loss in utility compared to uncompressed baselines.

## 1  Introduction

Federated Learning (FL) is a distributed machine learning (ML) paradigm that trains a model across a number of participating entities holding local data samples. In this work, we focus on *cross-device* FL that harnesses a large number (hundreds of millions) of edge devices with disparate characteristics such as availability, compute, memory, or connectivity resources (Kairouz et al., 2019).

Two challenges to the success of cross-device FL are privacy and scalability. FL was originally motivated for improving privacy since data points remain on client devices. However, as with other forms of ML, information about training data can be extracted via membership inference or reconstruction attacks on a trained model (Carlini et al., 2021a,b; Watson et al., 2022), or leaked through local updates (Melis et al., 2019; Geiping et al., 2020). Consequently, Secure Aggregation (SECAGG) protocols were introduced to prevent the server from directly observing individual client updates, which is a major vector for information leakage (Bonawitz et al., 2019). Additional mitigations such as Differential Privacy (DP) may be required to offer further protection against attacks (Dwork et al., 2006; Abadi et al., 2016), as discussed in Section 5.

Ensuring scalability to hundreds of populations of heterogeneous clients is the second challenge for FL. Indeed, wall-clock training times are highly correlated with increasing model and batch sizes (Huba et al., 2022), even with recent efforts such as FedBuff (Nguyen et al., 2022), and communication overhead between the server and clients dominates model convergence time. Consequently, compression techniques were used to reduce the communication bandwidth while maintaining model accuracy. However, a fundamental problem has been largely overlooked in the literature: in their native form, standard compression methods such as scalar quantization and pruning are not compatible with SECAGG. This makes it challenging to ensure both privacy and communication efficiency.

Submitted to 36th Conference on Neural Information Processing Systems (NeurIPS 2022). Do not distribute.

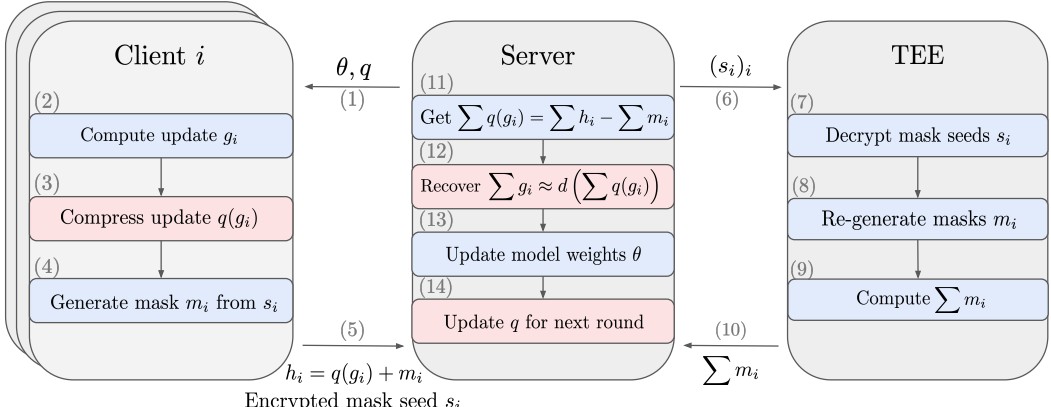

Figure 1: Summary of the proposed approach for one FL round, where we omit the round dependency and Differential Privacy (DP) for clarity. Blue boxes denote standard steps and red boxes denote additional steps for uplink compression. Client $i$ computes local model update $g_i$, compresses it with the compression operator $q$, and encrypts it by adding a random mask $m_i$ in the compressed domain, hence reducing the uplink bandwidth (steps 2–4). The server recovers the aggregate in the compressed domain by leveraging any SECAGG protocol (steps 7–13, with a TEE-based SECAGG). Since the decompression operator $d$ is linear, the server can convert the aggregate back to the non-compressed domain, up to compression error (step 12). As with the model weights $\theta$, the compression operator $q$ are also periodically updated and broadcast by the server (step 14). In Section 3, we apply the proposed method to scalar quantization and pruning without impacting SECAGG and propose Secure Indexing, a variant of SECAGG for extreme uplink compression with product quantization.

In this paper, we address this gap by adapting compression techniques to make them compatible with SECAGG. We focus on compressing *uplink* updates from clients to the server for three reasons. First, uplink communication is more sensitive and so is subject to a high security bar, whereas downlink updates broadcast by the server are deemed public. Second, upload bandwidth is generally more restricted than download bandwidth. For instance, according to the most recent FCC[1] report, the ratio of download to upload speeds for DSL and cable providers[2] in the US ranges between $3\times$ to $20\times$ (FCC, 2021). Finally, efficient uplink communication brings several benefits beyond speeding up convergence: lowering communication cost reduces selection bias due to under-sampling clients with limited connectivity, improving fairness and inclusiveness. It also shrinks the carbon footprint of FL, the fraction of which attributable to communication can reach 95% (Qiu et al., 2021).

In summary, we present the following contributions in this paper:

- We highlight the fundamental mismatch between two critical components of the FL stack: SECAGG protocols and uplink compression mechanisms.
- We formulate solutions by imposing a linearity constraint on the decompression operator, as illustrated in Figure 1 in the case of TEE-based SECAGG.
- We adapt the popular scalar quantization and (random) pruning compression methods for compatibility with the FL stack that require no changes to the SECAGG protocol.
- For extreme uplink compression without compromising security, we propose Secure Indexing (SECIND), a variant of SECAGG that supports product quantization.

## 2 Compression Techniques

In this subsection, we consider a matrix $W \in \mathbb{R}^{C_{\text{in}} \times C_{\text{out}}}$ representing the weights of a linear layer to introduce three major compression methods (scalar quantization, pruning and product quantization) with distinct compression/accuracy tradeoffs and identify the challenges SECAGG faces to be readily amenable to these popular compression algorithms. We discuss product quantization below and refer the reader to Appendix A.3 for a detailed description of scalar quantization and pruning approaches.

---

[1] US Federal Communications Commission.

[2] FL is typically restricted to using unmetered connections, usually over Wi-Fi (Huba et al., 2022).

**Product Quantization.** Product quantization (PQ) is a compression technique developed for nearest-neighbor search (Jégou et al., 2011) that can be applied for model compression (Stock et al., 2020). Here, we show how we can re-formulate PQ to represent model updates. We focus on linear layers and refer the reader to Stock et al. (2020) for adaptation to convolutions. Let the *block size* be $d$ (say, 8), the number of *codewords* be $k$ (say, 256) and assume that the number of input channels, $C_{\text{in}}$, is a multiple of $d$. To compress $W$ with PQ, we evenly split its columns into subvectors or blocks of size $d \times 1$ and learn a *codebook* via $k$-means to select the $k$ codewords used to represent the $C_{\text{in}} \times C_{\text{out}}/d$ blocks of $W$. PQ with block size $d = 1$ amounts to non-uniform scalar quantization with $\log_2 k$ bits per weight. The PQ-compressed matrix $W$ is represented with the tuple $(C, A)$, where $C$ is the codebook of size $k \times d$ and $A$ gives the assignments of size $C_{\text{in}} \times C_{\text{out}}/d$. Assignments are integers in $[0, k-1]$ and denote which codebook a subvector was assigned to. To decompress the matrix (up to reshaping), we index the codebook with the assignments, written in PyTorch-like notation as $\widehat{W} = C[A]$. There are several obstacles to making PQ compatible with SECAGG. First, each client may have a different codebook, and direct access to these codebooks is needed to decode each client's message. Even if all clients share a (public) codebook, the operation to take assignments to produce an (aggregated) update is not linear, and so cannot be directly wrapped inside SECAGG.

# 3   Method

In this section, we propose solutions to reconcile security (SECAGG) and communication efficiency. Our approach is to modify compression techniques to share some hyperparameters globally across all clients so that aggregation can be done by uniformly combining each client's response, while still ensuring that there is scope to achieve accurate compressed representations. As detailed below, each of the proposed methods offers the same level of security as standard SECAGG without compression.

## 3.1   Secure Aggregation and Compression

We propose to compress the uplink model updates $g_i$ through a compression operator $q$, whose parameters are round-dependent but the same for all clients participating in the same round. Then, we will add a random mask $m_i$ to each compressed client update $q(g_i)$ in the compressed domain, thus effectively reducing uplink bandwidth while ensuring that $h_i = q(g_i) + m_i$ is statistically indistinguishable from any other representable value in the finite group (see Appendix A.2). In this setting, SECAGG allows the server to recover the aggregate of the client model updates in the compressed domain: $\sum_i q(g_i)$. If the decompression operator $d$ is linear, the server is able to recover the aggregate in the non-compressed domain, up to compression error, as illustrated in Figure 1:

$$d\left(\sum_i h_i - \sum_i m_i\right) = d\left(\sum_i q(g_i)\right) = \sum_i d(q(g_i)) \approx \sum_i g_i.$$

The server periodically updates the compression and de-compression operator parameters, either from the aggregated model update, which is deemed public, or by emulating a client update on some similarly distributed public data. Once these parameters are updated, the server broadcasts them to the clients for the next round. This adds overhead to the downlink communication payload, however, this is negligible compared to the downlink model size to transmit. For instance, for scalar quantization, $q$ is entirely characterized by one `fp32` scale and one `int32` zero-point per layer, the latter of which is unnecessary in the case of a symmetric quantization scheme. Finally, this approach is compatible with both synchronous FL methods such as FedAvg (McMahan et al., 2017) and asynchronous methods such as FedBuff (Nguyen et al., 2022) as long as SECAGG maintains the mapping between the successive versions of quantization parameters and the corresponding client updates.

## 3.2   Application

Next, we show how we adapt scalar quantization and random pruning with no changes required to SECAGG. We illustrate our point with TEE-based SECAGG while these adapted uplink compression mechanisms are agnostic of the SECAGG mechanism. Finally, we show how to obtain extreme uplink compression by proposing a variant of SECAGG, which we call SECIND. This variant supports product quantization and is provably secure. In the following discussion, we refer the reader to Appendix A.2 for additional context related to SECAGG such as finite group sizes and mask seeds.

### 3.2.1 Scalar Quantization and Secure Aggregation

A model update matrix $g_i$ compressed with $b$-bit scalar quantization is given by an integer representation in the range $[0, 2^b - 1]$ and by the quantization parameters *scale* ($s$) and *zero-point* ($z$). A sufficient condition for the decompression operator to be linear is to broadcast common quantization parameters per layer for each client. Denote $q(g_i)$ as the integer representation of quantized client model update $g_i$ corresponding to a particular layer for client $1 \leq i \leq N$. Set the scale of the decompression operator to $s$ and its zero-point to $z/N$. The decompression operating on a quantized weight $w_q$ is linear given by $w_q \mapsto s \times (w_q - \frac{z}{N})$. Then, the server is able to decompress as follows:

$$d\left(\sum_i q(g_i)\right) = s \sum_i q(g_i) - \frac{z}{N} = \sum_i \left(s(q(g_i)) - z\right) \approx \sum_i g_i$$

Recall that all operations are performed in a finite group. Therefore, to avoid overflows at aggregation time, we quantize with a bit-width $b$ but take SECAGG bit-width $p > b$, thus creating a margin for potential overflows. This approach is related to the fixed-point aggregation described in (Bonawitz et al., 2019; Huba et al., 2022), but we calibrate the quantization parameters and perform the calibration per layer and periodically, unlike the related approaches.

**Privacy, Security and Bandwidth.** Scales and zero points are determined from public data on the server. Downlink overhead is negligible: the server broadcasts the per-layer quantization parameters. The upload bandwidth is $p$ bits per weight, where $p$ is the SECAGG finite group size. Since the masks $m_i$ are chosen in the integer range $[0, 2^p - 1]$, any masked integer representation taken modulo $2^p$ is statistically indistinguishable from any other vector.

### 3.2.2 Pruning and Secure Aggregation

To enable linear decompression with random pruning, all clients will share a common pruning mask for each round. This can be communicated compactly before each round as a seed for a pseudo-random function. This pruning mask seed is different from the SECAGG mask seed described in Appendix A.2 and has a distinct role. Each client uses the pruning seed to reconstruct a pruning mask, prunes their model update $g_i$, and only needs to encrypt and transmit the unpruned parameters. The trade-off here is that some parameters are completely unobserved in a given round, as opposed to traditional pruning. SECAGG operates as usual and the server receives the sum of the tensor of unpruned parameters computed by participating clients in the round, which it can expand using the mask seed. We denote the pruning operator as $\phi$ applied to the original model update $g_i$, and the decompression operator as $d$ applied to a compressed tensor $\phi(g_i)$. Decompression is an expansion operation equivalent to multiplication with a sparse permutation matrix $P_i$ whose entries are dependent on the $i$'th client's mask seed. Crucially, when all clients share the same mask seed within each round, we have $P_i = P$ for all $i$ and linearity of decompression is maintained:

$$d\left(\sum_i \phi(g_i)\right) = P\left(\sum_i \phi(g_i)\right) = \sum_i P_i \phi(g_i) = \sum_i d(\phi(g_i)) \approx \sum_i g_i.$$

**Privacy, Security and Bandwidth.** Since the mask is random, no information leaks from the pruning mask. The downlink overhead (the server broadcasts one integer mask seed) is negligible. The upload bandwidth is simply the size of the sparse client model updates. Finally, there is no loss in security since each client uses standard SECAGG mechanism on the non-pruned entries.

### 3.2.3 Product Quantization and Secure Indexing

We next describe the Secure Indexing (SECIND) primitive, and discuss how to instantiate it. Recall that with PQ, each layer has its own codebook $C$ as explained in Section 3. Let us fix one particular layer compressed with codebook $C$, containing $k$ codewords. We assume that $C$ is common to all clients participating in the round. Consider the assignment matrix of a given layer $(A^i)_{m,n}$ for client $i$. From these, we seek to build the *assignment histograms* $H_{m,n} \in \mathbb{R}^k$ that satisfy

$$H_{m,n}[r] = \sum_i \mathbf{1}\left(A^i_{m,n} = r\right),$$

---

**Algorithm 1** Secure Indexing (SECIND)

---

1: **procedure** SECUREINDEXING(C)                                    ▷ This happens inside the TEE
2:     Receive common codebook $C$ from server            ▷ $C$ is periodically updated by the server
3:     Initialize histograms $H_{m,n}$ to 0             ▷ Each histogram for block $(m,n)$ has size $k$
4:     **for** each client $i$ **do**
5:         Receive and decrypt assignment matrix $A^i$
6:         **for** each block index $(m,n)$ **do**
7:             $r \leftarrow A^i_{m,n}$                            ▷ Recover assignment of client $i$ for block $(m,m)$
8:             $H_{m,n}[r] \leftarrow H_{m,n}[r] + 1$                  ▷ Update global count for codeword index $r$
9:     Send back histograms $H_{m,n}$ to the server

---

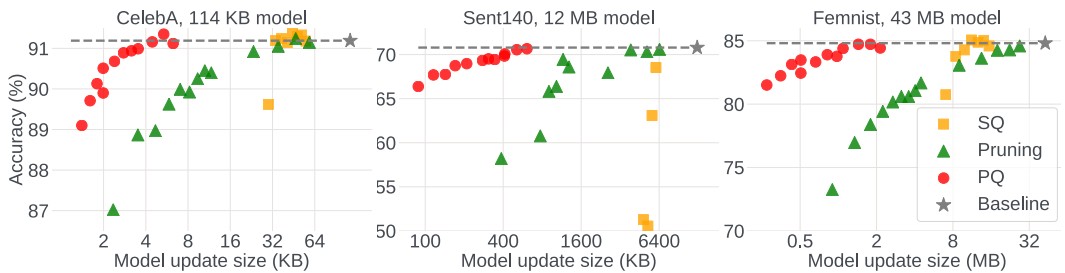

Figure 2: We adapt scalar quantization (SQ) and pruning to the SECAGG protocol to enable efficient and secure uplink communications. We also present results for product quantization (PQ) under the proposed novel SECIND protocol. *The $x$ axis is log-scale* and represents the uplink message size. Baseline refers to SECAGG FL run without any uplink compression, displayed as a horizontal line for easier comparison. Model size is indicated in the plot titles. Uncompressed client updates are as large as the models when $p = 32$ (see Appendix A.2, represented as stars).

where the indicator function $\mathbf{1}$ satisfies $\mathbf{1}\left(A^i_{m,n} = r\right) = 1$ if $A^i_{m,n} = r$ and $0$ otherwise. A *Secure Indexing* primitive will produce $H_{m,n}$ while ensuring that no other information about client assignments or partial aggregations is revealed. The server receives assignment histograms from SECIND and is able to recover the aggregated update for each block indexed by $(m,n)$ as $\sum_r H_{m,n}[r] \cdot C[r]$.

We describe how SECIND can be implemented with a TEE in Algorithm 1. Each client encrypts the assignment matrix, for instance with additive masking as described in Section A.2, and sends it to the TEE via the server. Hence, the server does not have access to the plaintexts client-specific assignments. TEE decrypts each assignment matrix and for each block indexed by $(m,n)$ produces the assignment histogram. Compared to SECAGG, where the TEE receives an encrypted seed per client (a few bytes per client) and sends back the sum of the masks $m_i$ (same size as the considered model), SECIND receives the (masked) assignment matrices and sends back histograms for each round. SECIND implementation feasibility is briefly discussed in Appendix A.7.

**Privacy, Security and Bandwidth.** Codebooks are computed from public data while individual assignments are never revealed to the server. The downlink overhead of sending the codebooks is negligible and for more details, please refer to Appendix A.6.3. The upload bandwidth in the TEE implementation is the assignment size, represented in $k$ bits (the number of codewords). For instance, with a block size $d = 8$ and $k = 32$ codewords, assignment storage costs are 5 bits per 8 weights, which converts to 0.625 bits per weight. The tradeoff compared to non-secure PQ is the restriction to a global codebook for all clients (instead of one tailored to each client), and the need to instantiate SECIND instead of SECAGG. Since the assignments are encrypted before being sent to the TEE, there is no loss in security. Here, any encryption mechanism (not necessarily relying on additive masking) would work.

## 4 Experiments

In this section, we numerically evaluate the performance of the proposed approaches when adapted to SECAGG protocols. We study the relationship between uplink compression and model accuracy

for the LEAF benchmark tasks. In addition, for scalar and product quantization we also analyze the impact of refresh rate for compression parameters on overall model performance.

## 4.1 Experimental Setup

We closely follow the setup of Nguyen et al. (2022) and use the FLSim library for our experiments . All experiments are run on a single V100 GPU 16 GB (except for Sent140 where we use one V100 32 GB) and typically take a few hours to run. More experiment details can be found in Appendix A.4.

**Tasks.** We run experiments on three datasets from the LEAF benchmark (Caldas et al., 2018): CelebA (Liu et al., 2015), Sent140 (Go et al., 2009) and FEMNIST (LeCun and Cortes, 2010). For CelebA, we train the same convolutional classifier as Nguyen et al. (2022) with BatchNorm layers replaced by GroupNorm layers and 9,343 clients. For Sent140, we train an LSTM classifier for binary sentiment analysis with $59,400$ clients. Finally, for FEMNIST, we train a GroupNorm version of the ResNet18 (He et al., 2016) for digit classification with 3,550 clients. For all compression methods, we do not compress biases and norm layers for their small overhead.

**Baselines.** We focus here on the (synchronous) FedAvg approach although, as explained in Section 3, the proposed compression methods can be readily adapted to asynchronous FL aggregation protocols. As done in the literature, we keep the number of clients per round to at most 100, a small fraction of the total considered population size (Chen et al., 2019; Charles et al., 2021). We report the average and standard deviation of accuracy over three independent runs for all tasks at different uplink byte sizes corresponding to various configurations of the compression operator.

**Implementation Details.** The downlink overhead of sending the per-layer codebooks for product quantization is negligible as shown in Appendix A.6.3. Finally, the convergence time in terms of rounds is similar for PQ runs and the non-compressed baseline, as illustrated in Appendix A.6.4. Note that outside a simulated environment, the wall-clock time convergence for PQ runs would be *lower* than the baseline since uplink communication would be more efficient, hence faster.

## 4.2 Results and Comparison with Prior Work

Results for efficient and secure uplink communications are displayed in Figure 2, where PQ yields a consistently better trade-off curve between model update size and accuracy. For instance, on CelebA, PQ achieves $\times 30$ compression with respect to the non-compressed baseline at iso-accuracy. The iso-accuracy compression rate is $\times 32$ on Sent140 and $\times 40$ on FEMNIST (see Appendix for detailed tables). Scalar quantization accuracy degrades significantly for larger compression rates due to the overflows at aggregation as detailed in Appendix A.6.1. Pruning gives intermediate tradeoffs between scalar quantization and product quantization. The line of work that develops FL compression techniques mainly includes FetchSGD (Rothchild et al., 2020) although the authors do not mention SECAGG. Their results are not directly comparable to ours due to non-matching experimental setups (e.g., datasets and architectures). However, Figure 6 in the appendix of Rothchild et al. (2020) mentions upload compression rates at iso-accuracy that are weaker than those obtained with PQ.

## 5 Conclusion

In this paper, we reconcile efficiency and security for uplink communication in Federated Learning. We propose to adapt existing compression mechanisms such as scalar quantization and pruning to the secure aggregation protocol by imposing a linearity constraint on the decompression operator. Our experiments demonstrate that we can adapt both quantization and pruning mechanisms to obtain a high degree of uplink compression with minimal degradation in performance and higher security guarantees. For achieving the highest rates of compression, we introduce SECIND, a variant of SECAGG well-suited for TEE-based implementation that supports product quantization while maintaining a high security bar. While our primary focus is on enabling efficient and secure uplink communication, our proposed approaches are compatible with user-level DP. For instance, DP noise can be added natively by the TEE with our modified random pruning or scalar quantization approaches. For PQ and SECIND, it would require, however, to transfer the aggregation to TEE or to design a DP mechanism in the assignment space, since DP noise must be added by the TEE and not by the server. For future work, we plan to investigate this further, and also extend our work to other federated learning scenarios such as asynchronous federated learning.

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

# A Appendix

## A.1 Related Work

Communication is identified as a primary efficiency bottleneck in FL, especially in the cross-device FL setting (Kairouz et al., 2019). This has led to significant interest in reducing FL's communication requirements. In what follows, we might refer to any local model update in a distributed training procedure as a *gradient*, including model updates computed following multiple local training steps.

**Bi-directional Compression.** In addition to uplink gradient compression, a line of work also focuses on downlink model compression. In a non-distributed setup, Zhou et al. (2016); Courbariaux et al. (2015) demonstrates that it is possible to meaningfully train with low bit-width models and gradients. In FL, Jiang et al. (2019) proposes adapting the model size to the device to reduce both communication and computation overhead. Since the local models are perturbed due to compression, researchers propose adapting the optimization algorithm for better convergence (Liu et al., 2020; Sattler et al., 2020; Tang et al., 2019; Zheng et al., 2019; Amiri et al., 2020; Philippenko and Dieuleveut, 2021). Finally, pre-conditioning models during FL training can allow for quantized on-device inference, as demonstrated for non-distributed training by Gupta et al. (2015); Krishnamoorthi (2018). As stated in Section 1, we do not focus on downlink model compression since uplink bandwidth is the main communication bottleneck and since SECAGG only involves uplink communication.

**Aggregation in the Compressed Domain.** In the distributed setting, Yu et al. (2018) propose to leverage both gradient compression and parallel aggregation by performing the *ring all-reduce* operation in the compressed domain and decompressing the aggregate. To do so, the authors exploit temporal correlations of the gradients to design a linear compression operator. Another method, PowerSGD (Vogels et al., 2019), leverages a fast low-rank gradient compressor. However, both aforementioned methods are not evaluated in the FL setup and do not mention SECAGG. Indeed, the proposed methods focus on decentralized communication between the workers by leveraging the all-reduce operation. Moreover, Power SGD incorporates (stateful) error feedback on all distributed nodes, which is not readily adaptable to cross-device FL in which clients generally participate in a few (not necessarily consecutive) rounds. Finally, Rothchild et al. (2020) proposes FetchSGD, a compression method relying on a CountSketch, which is compatible with SECAGG.

## A.2 Secure Aggregation

SECAGG refers to a class of protocols that allow the server to aggregate client updates without accessing them individually. While SECAGG alone does not entirely prevent client data leakage, it is a powerful and widely-used component of current at-scale cross-device FL implementations (Kairouz et al., 2019). Two main approaches exist in practice: software-based protocols relying on Multiparty Computation (MPC) (Bonawitz et al., 2019; Bell et al., 2020; Yang et al., 2022), and those that leverage hardware implementations of Trusted Execution Environments (TEEs) (Huba et al., 2022).

SECAGG relies on additive masking, where clients protect their model updates $g_i$ by adding a uniform random mask $m_i$ to it, guaranteeing that each client's masked update is statistically indistinguishable from any other value. At aggregation time, the protocol ensures that all the masks are canceled out. For instance, in an MPC-based SECAGG, the pairwise masks cancel out within the aggregation itself, since for every pair of users $i$ and $j$, after they agree on a matched pair of input perturbations, the masks $m_{i,j}$ and $m_{j,i}$ are constructed so that $m_{i,j} = -m_{j,i}$. Similarly and as illustrated in Fig. 1, in a TEE-based SECAGG, the server receives $h_i = g_i + m_i$ from each client as well as the sum of the masks $\sum_i m_i$ from the TEE and recovers the sum of the updates as

$$\sum_i g_i = \sum_i h_i - \sum_i m_i.$$

**DP noise.** Regarding the addition of DP noise, while our primary focus is on enabling efficient and secure uplink communication, we emphasize that the proposed approaches are compatible with user-level DP. For instance, at the cost of increasing the complexity of the trusted computing base, DP noise can be added natively by the TEE with our modified random pruning or scalar quantization approaches. For PQ and SECIND, we can have the TEE to add noise in the assignment space (i.e., outputting a noisy histogram), or to map the histogram to the codeword space and add noise there. Each option offers a different tradeoff between privacy, trust, and accuracy; we leave detailed evaluation to future

work. it would require, however, to transfer the aggregation to TEE or to design a DP mechanism in the assignment space, since DP noise must be added by the TEE and not by the server.

**Finite Group.** SECAGG requires that the plaintexts—client model updates—be elements of a finite group, while the inputs are real-valued vectors represented with floating-point types. This requirement is usually addressed by converting client updates to fixed-point integers and operating in a finite domain (modulo $2^p$) where $p$ is typically set in prior literature to 32 bits. The choice of SECAGG bit-width $p$ must balance communication costs with the accuracy loss due to rounding and overflows.

**Minimal Complexity.** TEE-based protocols offer greater flexibility in how individual client updates can be processed; however, the code executed inside TEE is part of the trusted computing base (TCB) for all clients. In particular, it means that this code must be stable, auditable, defects- and side-channel-free, which severely limits its complexity. Hence, in practice, we prefer compression techniques that are either oblivious to SECAGG's implementation or require minimal changes to the TCB.

### A.3 Compression Methods

#### A.3.1 Scalar Quantization

Uniform scalar quantization maps floating-point weight $w$ to $2^b$ evenly spaced bins, where $b$ is the number of bits. Given a floating-point scale $s > 0$ and an integer shift parameter $z$ called the zero-point, we map any floating-point parameter $w$ to its nearest bin indexed by $\{0, \ldots, 2^b - 1\}$:

$$w \mapsto \text{clamp}(\text{round}(w/s) + z, [0, 2^b - 1]).$$

The tuple $(s, z)$ is often referred to as the quantization parameters (`qparams`). With $b = 8$, we recover the popular `int8` quantization scheme (Jacob et al., 2018), while setting $b = 1$ yields the extreme case of binarization (Courbariaux et al., 2015). The quantization parameters $s$ and $z$ are usually calibrated after training a model with floating-point weights using the minimum and maximum values of each layer. The compressed representation of weights $W$ consists of the `qparams` and the integer representation matrix $W_q$ where each entry is stored in $b$ bits. Decompressing any integer entry $w_q$ of $W_q$ back to floating point is performed by applying the (linear) operator $w_q \mapsto s \times (w_q - z)$.

**Challenge.** The discrete domain of quantized values and the finite group required by SECAGG are not natively compatible because of the overflows that may occur at aggregation time. For instance, consider the extreme case of binary quantization, where each value is replaced by a bit. We can represent these bits in SECAGG with $p = 1$, but the aggregation will inevitably result in overflows.

#### A.3.2 Pruning

Pruning is a class of methods that remove parts of a model such as connections or neurons according to some pruning criterion, such as weight magnitude (Le Cun et al. (1989); Hassibi and Stork (1992); see Blalock et al. (2020) for a survey). Konečný et al. (2016) demonstrate client update compression with random sparsity for federated learning. Motivated by previous work and the fact that random masks do not leak information about the data on client devices, we will leverage random pruning of client updates in the remainder of this paper. A standard method to store a sparse matrix is the coordinate list (COO) format[3], where only the non-zero entries are stored (in floating point or lower precision), along with their integer coordinates in the matrix. This format is compact, but only for a large enough compression ratio, as we store additional values for each non-zero entry. Decompression is performed by re-instantiating the uncompressed matrix with both sparse and non-sparse entries.

**Challenge.** Pruning model updates on the client side is an effective compression approach as investigated in previous work. However, the underlying assumption is that clients have different masks, either due to their seeds or dependency on client update parameters (*e.g.* weight magnitudes). This is a challenge for SECAGG as aggregation assumes a dense compressed tensor, which is not possible to construct when the coordinates of non-zero entries are not the same for all clients.

### A.4 Experimental Details

In this section, we provide further details of the experimental setup described in Section 4.1 and the hyper-parameters used for all the runs in Table 1. For all the tasks, we use a mini-batch SGD

---

[3]See the torch.sparse documentation.

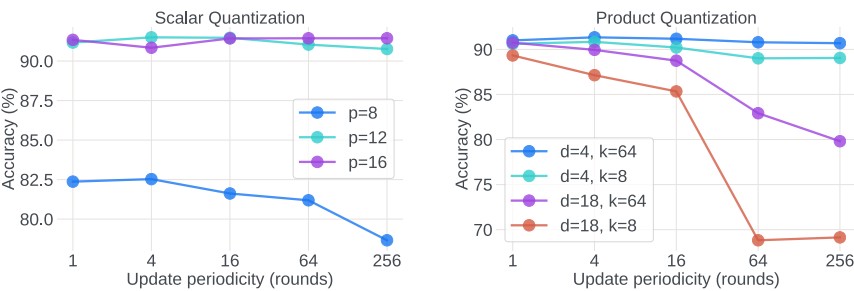

Figure 3: Impact of the refresh rate of the compression operator by the server on the CelebA dataset. **Left**: for scalar quantization (quantization parameters), where we fix the quantization bit-width $b = 8$ ($p$ denotes the SECAGG bit-width). **Right**: for product quantization (codebooks), where $k$ denotes the number of codewords and $d$ the block size.

optimizer for local training at the client and FEDAVG optimizer for global model update on the server. The LEAF benchmark is released under the BSD 2-Clause License.

**Baselines.** We run hyper-parameter sweeps to tune the client and server learning rates for the uncompressed baselines. Then, we keep the same hyper-parameters in all the runs involving uplink compression. We have observed that tuning the hyper-parameters for each compression factor does not provide significantly different results than using those for the uncompressed baselines, in addition to the high cost of model training involved.

**Compression details.** For scalar quantization, we use per-tensor quantization with MinMax observers. We use the symmetric quantization scheme over the integer range $[-2^{b-1}, 2^{b-1} - 1]$. For pruning, we compute the random mask separately for each tensor, ensuring all pruned layers have the same target sparsity in their individual updates. For product quantization, we explore various configurations by choosing the number of codewords per layer $k$ in $\{8, 16, 32, 64\}$ and the block size $d$ in $\{4, 9, 18\}$. We automatically adapt the block size for each layer to be the largest allowed one that divides $C_{\text{in}}$ (in the fully connected case).

Table 1: Hyper-parameters used for all the experiments including baselines. $\eta$ is the learning rate.

| Dataset | Users per round | Client epochs | Max. server epochs | $\eta_{\text{SGD}}$ | $\eta_{\text{FedAvg}}$ |
|---|---|---|---|---|---|
| CelebA | 100 | 1 | 30 | 0.90 | 0.08 |
| Sent140 | 100 | 1 | 10 | 5.75 | 0.24 |
| FEMNIST | 5 | 1 | 5 | 0.01 | 0.24 |

## A.5 Experimental Results

We provide various additional experimental results that are referred to in the main paper.

## A.6 Ablation Studies

We investigate the influence of the frequency of updates of the compression operator $q$ for scalar quantization and pruning, and study the influence of the SECAGG bit-width $p$ on the number of overflows for scalar quantization.

**Update frequency of the compression operators.** In Figure 3, we show that for scalar quantization, the update periodicity only plays a role with low SECAGG bit-width values $p$ compared to the quantization bit-width $b$. For product quantization, the update periodicity plays an important role for aggressive compression setups corresponding to large block sizes $d$ or to a smaller number of codewords $k$. For pruning, we measure the impact of masks that are refreshed periodically. We observe that if we refresh the compression operator more frequently, staleness is reduced, leading to accuracy improvements. We present our findings in Appendix A.6.5.

**Overflows for scalar quantization.** As discussed in Section 3.2.1, we choose the SECAGG bit-width $p$ to be greater than quantization bit-width $b$ in order to avoid aggregation overflows. While it suffices to set $p$ to be $\lceil \log_2 n_c \rceil$ more than $b$, where $n_c$ is the number of clients participating in the round, reducing $p$ is desirable to reduce uplink size. We study the impact of $p$ on the percentage of parameters that suffer overflows and present our findings in Appendix A.6.1.

### A.6.1   Aggregation overflows with Scalar Quantization

We discussed the challenge of aggregation overflows of quantized values with restricted SECAGG finite group size in Section A.3.1 and noted in Section 3.2.1 that it suffices for SECAGG bit-width $p$ to be greater than quantization bit-width $b$ by at most $\lceil \log_2 N \rceil$, where $N$ is the number of clients participating in a given round. However, the overflow margin increases the client update size by $p - b$ per weight. To optimize this further, we explore the impact of $p$ on aggregation overflows and accuracy, and present the results in Table 2. As expected, we observe a decrease in percentage of weights that overflow during aggregation with the increase in the overflow margin size. However, while there is some benefit to non-zero overflow margin size, there is no strong correlation between the overflow margin size and accuracy, indicating the potential to achieve better utility even in the presence of overflows.

Table 2: Percentage of aggregation overflows (among all model parameters) for the CelebA dataset over various SQ configurations. $b$ is Quantization bit-width, $p$ is SECAGG bit-width, $p - b$ is overflow margin size in bits.

| $b$ | $p$ | $p - b$ | Overflows (% of parameters) | Accuracy |
|---|---|---|---|---|
| 4 | 4 | 0 | 3.71±1.53 | 49.33±2.03 |
| 4 | 5 | 1 | 1.43±0.55 | 50.44±1.77 |
| 4 | 6 | 2 | 0.68±0.43 | 49.67±1.56 |
| 4 | 7 | 3 | 0.17±0.12 | 51.58±0.66 |
| 4 | 8 | 4 | 0.06±0.00 | 87.30±0.36 |
| 4 | 9 | 5 | 0.06±0.00 | 89.19±0.20 |
| 4 | 10 | 6 | 0.06±0.00 | 88.52±0.07 |
| 4 | 11 | 7 | 0.05±0.00 | 87.68±1.24 |
| 8 | 8 | 0 | 2.28±0.11 | 82.11±0.90 |
| 8 | 9 | 1 | 1.06±0.06 | 90.49±0.27 |
| 8 | 10 | 2 | 0.39±0.04 | 90.97±0.50 |
| 8 | 11 | 3 | 0.14±0.01 | 91.08±0.45 |
| 8 | 12 | 4 | 0.06±0.00 | 91.29±0.13 |
| 8 | 13 | 5 | 0.04±0.00 | 90.49±0.93 |
| 8 | 14 | 6 | 0.02±0.00 | 91.31±0.24 |
| 8 | 15 | 7 | 0.01±0.00 | 91.19±0.33 |

### A.6.2   Weighted aggregation and Scalar Quantization

Following the setup of Nguyen et al. (2022), we weight each client update by the number of samples the client trained on. Denoting the weight associated with the client $i$ with $\omega_i$ and following the same notations as in Section 3.1, weighted update is obtained as $h_i = (q(g_i) \times \omega_i) + m_i$. Since this is a synchronous FL setup, we do not set staleness factor. This weighted aggregation has no impact on pruning and product quantization, but can lead to overflows with scalar quantization. Therefore, we skip the weighting of quantized parameters of client updates and only weight non-quantized parameters (such as bias). For completion, we study with unweighted aggregation of client updates (including bias parameters) for scalar quantization experiments and present the result in Table **??**. As expected, these results are similar to the ones with weighted aggregation.

### A.6.3   PQ Codebook Size is Negligible

We demonstrate in Table 3 that the overhead of sending codebooks (for all layers) is negligible compared to the model size. When the model is very small (CelebA model is 114 KB), reducing $k$ and $d$ makes the overhead negligible without hurting performance.

Table 3: Cost of broadcasting codebooks (for downlink communications) is negligible compared to model sizes. Recall that $k$ denotes the number of codebooks and $d$ the block size.

| Dataset | Codebook size $k$ | Block size $d$ | Codebooks size (% of model size) |
|---|---|---|---|
| CelebA | 8 | 4 | 0.6 KB (0.5%) |
| | 8 | 18 | 2.5 KB (2.2%) |
| | 64 | 4 | 4.2 KB (3.7%) |
| | 64 | 18 | 14.6 KB (12.8%) |
| Sent140 | 8 | 4 | 0.9 KB (0.0%) |
| | 8 | 18 | 2.3 KB (0.0%) |
| | 64 | 4 | 5.4 KB (0.0%) |
| | 64 | 18 | 15.4 KB (0.1%) |
| FEMNIST | 8 | 4 | 2.6 KB (0.0%) |
| | 8 | 18 | 11.2 KB (0.0%) |
| | 64 | 4 | 20.8 KB (0.0%) |
| | 64 | 18 | 89.8 KB (0.2%) |

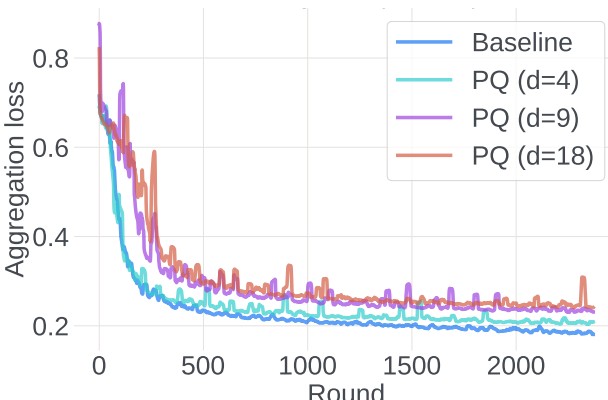

Figure 4: Number of rounds to convergence is similar for PQ-compressed runs compared to the non-compressed baseline (on CelebA). Note that outside a simulated environment, the wall-clock time convergence for PQ runs would be lower than the baseline since uplink communications would be faster.

### A.6.4 Convergence Curves

We also provide convergence curves for PQ-compressed and baseline runs to demonstrate similar number of rounds needed to convergence in Figure 4.

### A.6.5 Performance impact of sparsity mask refresh

In addition to scalar and product quantization as described in Section A.6, we also conduct experiments with varying the interval for refreshing pruning masks. We consider two levels of sparsity, 50% and 99% and our experiments are on the CelebA dataset. We present our results in Figure 5. Overall we find that the model accuracy is robust to the update periodicity unless at very high sparsities, where accuracy decreases when mask refresh periodicity increases. This is important for future directions such as in asynchronous FL where clients have to maintain the same mask across successive global updates.

### A.7 SECIND Implementations

SECIND can be extended to other settings, such as multi-party computation (using two or more servers to operate on shares of the input), where each client can send evaluations of *distributed point functions* to encode each assignment (Boyle et al., 2016). These are represented compactly, but

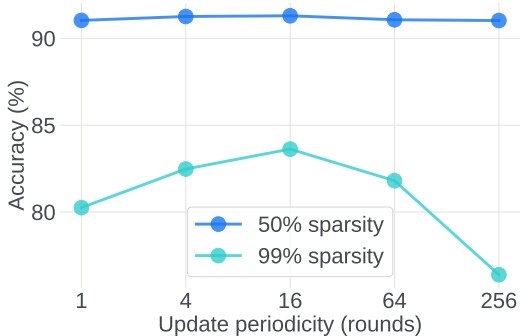

Figure 5: Impact of pruning mask refresh intervals on model performance for the CelebA dataset. Note that the effect of refreshing the pruning masks is more apparent at higher sparsity levels, and generalization performance decreases when masks are stale for longer during training.

may require longer codewords to overcome the overheads. We leave the study of such software implementations of SECIND to future work.