# OpenReview forum: "Reconciling Security and Communication Efficiency in Federated Learning"
_NeurIPS.cc/2022/Workshop/Federated_Learning — FL-NeurIPS 2022 Poster_

### Official Review · Reviewer_sXR9 · 2022-10-14
**Interesting paper combining compression with secure aggregation but leak detailed comparison with many related work**

In this paper, the authors investigated how to combine several uplink compression schemes with TEE in federated learning. They also provide preliminary numerical results to demonstrate the performance of their proposed framework. The authors studied an important research problem in FL, however, there is some related work (see below list) that combines compression schemes with a certain type of secure aggregation protocols, e.g., homomorphic encryption and secret sharing.  It is unclear what are the advantages of proposing schemes over the following scheme. Moreover, there is no formal security analysis or convergence analysis of the proposed schemes.

> Hosseini, Erfan, and Ashish Khisti. "Secure aggregation in federated learning via multiparty homomorphic encryption." 2021 IEEE Globecom Workshops (GC Wkshps). IEEE, 2021.

> Beguier, Constance, Mathieu Andreux, and Eric W. Tramel. "Efficient sparse secure aggregation for federated learning." arXiv preprint arXiv:2007.14861 (2020).

> Elkordy, Ahmed Roushdy, and A. Salman Avestimehr. "Secure aggregation with heterogeneous quantization in federated learning." arXiv preprint arXiv:2009.14388 (2020).

---

### Official Review · Reviewer_f9qw · 2022-10-17
**The proposed methods are trivial and need to be improved.**

This paper proposes adapt scalar quantization and pruning methods to secure aggregation to reduce the uplink communication cost.

The modification and adaption compression methods of existing scalar quantization and pruning is trivial, e.g. through sharing some hyperparameters globally. How to accurately determine or choose the mask? The server chooses it randomly and distributes to all clients, but whether the masks of different users are different, and the mask can be designed to make the compression ratio higher?

The paper proposes three compression methods, what are the differences and connections between them, and what are the differences in the specific use scenarios?

The experiment results are not sufficient to prove their conclusions.

---

### Official Review · Reviewer_BSf7 · 2022-10-18
**Paper 25 review**

The paper studies an important problem on the intersection of security and communication efficiency of federated learning. It suggest a way to efficiently combine quantization for uplink communication with secure aggregation.

The idea is quite clear and not hard to follow. The contribution is important since not many works try to tackle several federated learning issues of this kind. I believe that it can be interesting for the community.

Experimental results seem to be solid and comprehensive enough.

To conclude, I recommend acceptance to the workshop.

---

### Decision · Program_Chairs · 2022-10-20

Accept (Poster)